# Equivalent Circuit Models: An Effective Tool to Simulate Electric/Dielectric Properties of Ores—An Example Using Granite

**DOI:** 10.3390/ma15134549

**Published:** 2022-06-28

**Authors:** Kyosuke Fukushima, Mahmudul Kabir, Kensuke Kanda, Naoko Obara, Mayuko Fukuyama, Akira Otsuki

**Affiliations:** 1Graduate School of Engineering Science, Akita University, 1-1 Tegata Gakuen Machi, Akita 010-8502, Japan; m8020422@s.akita-u.ac.jp (K.F.); m8021405@s.akita-u.ac.jp (K.K.); obara@gipc.akita-u.ac.jp (N.O.); 2Graduate School of Engineering Science, Cooperative Major in Life Cycle Design Engineering, Tegata Campus, Akita University, 1-1 Tegata Gakuen Machi, Akita 010-8502, Japan; mayuko@gipc.akita-u.ac.jp; 3Ecole Nationale Supérieure de Géologie, GeoRessources, UMR 7359 CNRS, University of Lorraine, 2 Rue du Doyen, Marcel Roubault, BP 10162, 54505 Vandoeuvre-lès-Nancy, France; akira.otsuki@uai.cl; 4Facultad de Ingeniería y Ciencias, Universidad Adolfo Ibáñez, Diagonal Las Torres 2640, Peñalolén, Santiago 7941169, Chile; 5Waste Science & Technology, Luleå University of Technology, SE 971 87 Luleå, Sweden

**Keywords:** conductivity, dielectric constant, hard rock, mineral distribution, voltage-dependent resistance (VDR)

## Abstract

The equivalent circuit model is widely used in high-voltage (HV) engineering to simulate the behavior of HV applications for insulation/dielectric materials. In this study, equivalent circuit models were prepared in order to represent the electric and dielectric properties of minerals and voids in a granite rock sample. The HV electric-pulse application shows a good possibility of achieving a high energy efficiency with the size reduction and selective liberation of minerals from rocks. The electric and dielectric properties were first measured, and the mineral compositions were also determined by using a micro-X-ray fluorescence spectrometer. Ten patterns of equivalent circuit models were then prepared after considering the mineral distribution in granite. Hard rocks, as well as minerals, are dielectric materials that can be represented as resistors and capacitors in parallel connections. The values of the electric circuit parameters were determined from the known electric and dielectric parameters of the minerals in granite. The average calculated data of the electric properties of granite agreed with the measured data. The conductivity values were 53.5 pS/m (measurement) and 36.2 pS/m (simulation) in this work. Although there were some differences between the measured and calculated data of dielectric loss (*tanδ*), their trend as a function of frequency agreed. Even though our study specifically dealt with granite, the developed equivalent circuit model can be applied to any other rock.

## 1. Introduction

The modeling of rocks is one of the effective tools available to understand the physical phenomena in rocks under different applications, including the electric-pulse liberation of minerals. The electric-pulse liberation of minerals is an emerging comminution method that can be a good solution for low-energy efficiency in size reduction and selective liberation [1,2]. As rocks are dielectric materials [3,4,5,6], it is necessary to understand their electric and dielectric properties to evaluate and better understand the physics and behaviors of minerals upon the application of a HV electric pulse on them. Since the 1990s, computer simulations have become an important tool to model the research due to the unprecedented development of the data processing power of personal computers. There are many works related to electric-pulse liberation being carried out using computer simulations in order to understand the HV electric-pulse liberation of rocks [7].

Andres et al. (2001) compared the electrical pulse application with the conventional mechanical comminution of oxide ores containing hematite or platinum-group metals [1]. They also showed some simulation works of the electric-field distribution in ores using the electromagnetic theory of dielectric materials.

Some other works were performed to understand the electric-pulse comminution by using the COMSOL Multiphysics software package [8,9]. Seyed et al. (2015) worked on phosphate ore that was under electric-pulse application, which showed that the electrical field was dependent on the electrical properties of minerals, particle size and the location of conductive minerals [8]. Li et al. (2018) used the COMSOL Multiphysics software to understand the behavior of electro-pulse boring in granite under a HV application, considering the composition of granite, electrode spacing and electrode shape. They found that HV boring is affected by the composition of granite and its electric properties. However, they used equivalent circuit models of an electric pulse of a HV source only [9].

Zuo et al. (2015) discussed high-voltage pulse (HVP) breakage models of ores with three breakage indices (i.e., body breakage probability, body reduction evaluation index, body breakage product pre-weakening degree) [10]. They considered the breakage results of three ore samples (i.e., gold–copper ore, iron oxide copper–gold ore (IOCG), and hematite ore) and fitted the measured data statistically. Their work linked these parameters with the mass-specific impact energy, and even though the ores were different, the HVP model showed similar results with measured data [10].

Walsh and his groups conducted some simulation works related to the HV pulse liberation of ores [11,12], using Voronoi tessellation in order to simulate the mineral distribution in the granite rocks. Their works considered the minerals of granite to understand their behaviors under HV electric applications. However, these simulation works did not consider both the electric and dielectric properties of each mineral consisting of rock.

In HV engineering, equivalent circuit models that represent insulation and dielectric materials are used to simulate and understand the behavior of HV in testing objects [9,13,14,15,16,17]. Zuo et al. (2020) modeled insulation cable with equivalent circuit models by considering placing the cable into small fractions of insulation (dielectric) material and discussed the HV direct current influence on insulation cables [14]. Kabir et al. (2011) [18] made equivalent circuit models to understand the fine ceramics (ZnO varistors) by considering hundreds of ZnO grains of an average grain size of 1–2 μm to understand the nonlinear *I*–*V* (current–voltage) properties of ZnO varistors. Ono et al. (2009) [17] calculated the electric-field distribution in composite materials using equivalent circuit models under HV applications. Their works discussed different types of electric pulses and electrode gaps to understand HV behavior. With the above literature, the importance of considering each mineral to understand mineral liberation under a HV application is clearly identified. Additionally, it is worth noting that the equivalent circuit model analysis has a strong potential to understand the electric pulse comminution [18].

The equivalent circuit model is used in HV applications, as discussed before. The model can simulate the behavior of the subject under a HV application effectively, but only if the equivalent circuit can imitate the subject’s electrical/dielectric properties accurately. Thus, it is necessary to confirm the reproduction ability of the electrical/dielectric properties of a subject. However, the dielectric properties of composite materials are difficult to simulate by equivalent circuit models, as they are frequency-dependent and the dielectric values are fixed in an equivalent circuit model. We will discuss this regarding the simulation results.

We discussed the HV application on granite in the other paper [18], where an equivalent circuit was used to simulate the behavior of granite under a HV impulse application. As the results proved the effectiveness of our equivalent circuit model in electric-pulse liberation, in the current work, we aimed to further advance this model by considering voids’ (pores in minerals as well as rocks) behavior under a HV impulse on the rocks by using the measurement data of electrical properties of the dielectric breakdown of air. Our previous paper also dealt with voids (pores in minerals), but we assumed those voids were filled with liquid, as the electrical disintegration (ED) method applies HV impulses in liquid. Thus, the values of electric resistance that include voids, did not represent the dielectric-breakdown properties of voids (pores) in minerals; rather, it showed the electric resistance of minerals including voids, as they were assumed to be filled with liquid. Generally, the electric and dielectric properties of any dielectric and insulation materials are measured in the HV branch in air [13]. In this paper, we considered all the voids (pores in minerals), and their equivalent circuits were placed along with the circuits of minerals in the equivalent circuit model of granite. Again, we considered both the electric and dielectric properties of each type of mineral of granite (i.e., quartz, plagioclase, K-feldspar, and biotite) and made an equivalent circuit to simulate the granite’s electric and dielectric properties. First, the electric and dielectric properties of the granite sample and the modal mineral composition of the sample were measured. Considering the granite ore is composed of the minerals and the voids, we calculated their values of electric resistance and capacitance and then created the circuit models to simulate the granite rock. For the simulation works, we prepared 10 types of mineral distributions randomly, and an equivalent circuit model was prepared for each mineral distribution. The simulation results of our works showed good agreement with the measured data of *I*–*V* properties and dielectric properties.

## 2. Materials and Methods

### 2.1. Sample

Our method of using an equivalent circuit to simulate rocks can be applied to any type of rock. We selected granite as an example of hard rock for this study. Our sample granite was bought from Kenis, Japan, and the sample was collected from Akaiwa city of Okayama prefecture, Japan. The sample granite was thinly sliced (with a thickness of about 1.1 mm) and its surface was well polished for measurements, including the ones for the elemental compositions as well as the electrical and dielectric properties. Figure 1a shows a snapshot of the granite sample. The elemental mapping was obtained by using a micro-X-ray fluorescence spectrometer (µXRF, M4 Tornedo Plus; Bruker, Billerica, MA, USA) and the mineral composition was then calculated by stoichiometry. Based on the imaging and x-ray analysis, whose results are shown in Figure 1b, the modal composition of the granite sample was calculated. This granite sample consists of quartz (30.1%), plagioclase (37.3%), K-feldspar (23.3%), and biotite (9.2%). The modal composition indicates that the granite is a monzogranite [19]. The modal composition was calculated, considering the composition of the abovementioned four minerals, but only without counting the porosity of the granite sample. In addition, the value of porosity (void% in rock) was needed for our simulation work and we obtained that information from the literature, which will be discussed in Section 2.2.

### 2.2. Experimental

The electric current–voltage (*I*–*V*) properties of our granite sample were measured. Granite is a well-known dielectric material [3,4,5,6] whose resistivity is quite large (10^10^ Ω·m [20]). Thus, when measuring the electric properties of granite, the surface leakage current should be avoided. The sample was placed in a resistivity chamber (Model 24, ADCMT, Saitama, Japan) where the leakage current of the sample was eliminated by a guard electrode of the chamber. The diameter of the upper electrode of this chamber was 6 mm. An ultra-high resistance meter (5451, ADCMT) was used to measure the *I*–*V* properties of the sample. In order to measure the dielectric properties of the sample, we used an LCR meter (ZM2376, NF, Yokohama, Japan) with the same resistivity chamber. The capacitance *C* and dielectric loss *tanδ* were measured with the range of frequency from 1 Hz to 1 MHz with 500 measurement points of frequency.

Any kind of rock contains voids (pores), though their volumes vary with minerals inside the rocks. Voids are not seen in the modal compositions of our granite sample from µXRF mapping, but we can assume their proportions from the literature [11]. Voids’ electrical characteristics can be understood by the gas discharge phenomena [13]. Air is one of the dielectric materials whose dielectric breakdown under a HV application is a large branch of interest in HV engineering [13,19]. In order to understand and simulate the behavior of voids in rocks, we also measured the *I*–*V* properties of the gas discharge by the same experimental procedure mentioned above.

### 2.3. Simulation Methods

In this study, the equivalent circuit model was used for simulation. An equivalent circuit model can simulate any kind of rock, however, here, we will discuss granite. The granite was assumed to be a cube of 1 mm edge length, and the model was composed of 1000 of 0.1 mm × 0.1 mm × 0.1 mm smaller cubes (Figure 2a). The model was created by assigning the minerals and voids to those smaller cubes of a 0.1mm edge length and making up the granite. The modal mineral composition of the granite sample discussed in Section 2.1 was used for the simulation. The porosity (i.e., volume % of voids) varies depending on the type and origin of the minerals as well as rocks. For granite, the literature reported that the porosity was from 0.9% to 2.6% [21,22]. In our work, the equivalent circuit model was created by assuming that the porosity in the granite was 2 vol%. The electrical properties of the granite were calculated by simulating the electrical properties of the minerals in the granite, and the circuit parameters were calculated for the equivalent circuit models. A circuit-preparing software program written in C# was developed to create a circuit file in the Netlist format, and then the circuit simulations were performed by using the LTspice circuit simulator.

The possible combinations of mineral distributions can be thousands of patterns. We generated 10 patterns by using random distributions that can decide which minerals or voids will be placed in among the 1000 cubes (0.1 mm edge length). First, void-equivalent circuit elements were placed in the divided cubes with a probability of 2%, and then mineral-equivalent circuit elements for a void were randomly placed in the remaining cubes with the ratio of minerals determined by our µXRF measurement (Figure 1b), based on the fact that the four minerals considered in granite have very similar specific gravities (i.e., quartz (2.65 [11]), plagioclase (2.56 [11]), K-feldspar (2.63 [11]), biotite (2.7–3.4 [23])). Figure 3 shows an example of a mineral distribution using the software of our simulation program. Figure 3a is the 3D model of the minerals and voids (pores) and Figure 3b is the XY plane image of the granite. As mentioned above, the placement of the minerals and voids was selected randomly by our software. When the distribution of the minerals and pores/voids were settled by our software, an equivalent circuit was created automatically to represent the granite sample with the determined distributions of minerals and voids.

## 3. Results and Discussion

### 3.1. Electrical Properties

Following the instructions of the ultra-high resistance meter, we started to take the data of the *I*–*V* properties from the input voltage of 300 V. The applied voltage was then increased from 300 V to 1000 V with an increment of 100 V. The measurements were carried out with an automatic measurement system developed by the LabVIEW software. The data were taken five times under each voltage and their average values and standard deviations were calculated. The relationship of *J*–*E* was calculated from the measured data of the *I*–*V* properties (see Figure 4). Here, *J* represents the electric current density and *E* represents the electric field calculated from the input voltage divided by the distance between the electrodes (i.e., the thickness of the sample, 1.1 mm). *J* was calculated by dividing the electric current *I* with the surface area with the upper electrode, whose diameter was 6 mm. Figure 4 is a double logarithmic graph. The relationship between *J* and *E* can be expressed in the following equation [13]:(1)J=σE
where *σ* is conductivity and the slope of the *J*–*E* relationship indicates the conductivity *σ* of the sample. The average value of conductivity was found to be 9.6 × 10^−11^ S/m and its standard deviation was 9.0 × 10^−12^ S/m. Thus, the value of resistivity (1/*σ*) of our sample was 1.0 × 10^10^ Ω·m, and it was similar to the value found in the literature (10^10^ Ω·m) [20].

### 3.2. Dielectric Properties of Granite

The capacitance *C* and dielectric loss *tanδ* were measured with the range of frequency described in Section 2.1 (1 Hz–1 MHz). The measurements were carried out with 500 measurement points over this frequency range. They were measured five times at each frequency and their average and standard deviation values were calculated. The measured data were plotted in double logarithmic graphs (Figure 5). The bars indicate the standard deviation of the values (average data ± standard deviation value). As shown in Figure 5b, the values of *C* decreased with frequency until 1000 Hz but they became almost constant from 1000 to 10,000 Hz. In the latter range, the average value of *C* was 2.45 pF and its standard deviation was 0.253 pF. There were some peak values found in the *C*–*f* properties. These types of peaks in capacitance *C*, as well as the dielectric constant, were found in the dielectric materials in a different range of frequencies, indicating dielectric relaxation [24]. Dielectric relaxation occurs in composite/polymer materials. Dielectric polarization occurs in dielectric materials when a voltage is applied to them. For an AC source of voltage, the re-orientational motions of dipoles and the translational motions of a charged area differ for different types of materials or polymer chains in a composite. This difference, seen at the re-orientational motion, appeared in a resonance form with some frequencies and, hence, the dielectric relaxation occurred in the composite/polymer materials. It is to be noted, however, that the imaginary part of the permittivity shows the dielectric relaxation more clearly [24], but as it is not a matter of discussion for this paper, we have avoided the details here. Peak values were also found in *tanδ*. The average value of *tanδ* from the range of 1000 to 10,000 Hz was 0.773 and the average standard deviation was 0.151. The dielectric constant *ε* was calculated from the Equation (2) [13,25]:(2)$$C=\varepsilon \frac{S}{L}\;$$
where *C* is the capacitance, *ε* (*ε* = *ε*_r_*ε*_0_) is the dielectric constant, *L* is the thickness of the sample (1.1 mm) and *S* is the surface area of the electrode (28.3 × 10^−6^ m^2^). *Ε*_r_ is the relative dielectric constant/permittivity of granite (i.e., 4–7 at 100 MHz) [26,27] and ε_0_ is the dielectric constant of vacuum/air (8.85 × 10^−12^ F/m). From our measurements, the average value of *ε*_r_ was found to be 10.8 at the range of frequency from 1000 to 10,000 Hz (10 kHz). The value was not exactly the same as the literature values (i.e., 4–7 at 100 MHz) [26,27]. However, it should be noted that the frequency used in the above reference was 100 MHz, and in general, the capacitance *C* and the relative dielectric constant *ε*_r_ decreased with frequency. Using the calculated values of *ε*, the *ε*-*f* properties of our granite sample were plotted in a double logarithmic graph (Figure 6). The resistance *R* is expressed by the following equation, which was used to calculate the general resistance value [25]:(3)$$R=\frac{1}{\sigma }\frac{L}{S}$$
where *R* is resistance, *σ* is the electrical conductivity, *L* is length (i.e., 1.1 mm), and *S* is the cross-sectional area (i.e., the surface area of the upper electrode, 28.3 × 10^−6^ m^2^). The problem of an electromagnetic wave in dielectric materials (conductivity *σ* ≠ 0) can be understood by solving Maxwell’s equations for electromagnetics [25]. We will not discuss the details here, but by solving Maxwell’s equations for dielectric materials, the following relationship can be found [25]:(4)$$tan\delta =\frac{\sigma }{\omega \varepsilon }$$
where *tanδ* is the dielectric loss, *ε* is the dielectric constant and *ω* represents the angular frequency (i.e., *ω* = 2*π**f* where *f* is frequency). Taking the logarithm of both sides of this equation, we obtain Equation (5) [25],
(5)logtanδ =logσ−logω−logε

This equation reflects the behavior of the dielectric materials under the electromagnetic wave, though it does not reflect the dielectric relaxation phenomenon [25,26]. Due to the polarization in a dielectric material (i.e., granite), the change in the dielectric constant/permittivity depends on the frequency *f* of the applied electric field. There were some peaks in the dielectric properties (i.e., *C*: 1.68 × 10^−12^ F (393 Hz), 8.44 × 10^−13^ F (21 kHz), 1.59 × 10^−12^ F (435 kHz), 4.37 × 10^−13^ F (761 kHz)—Figure 5a; *tanδ*:0.109 (18.2 kHz), 0.479 (33.4 kHz), 0.0682 (533 kHz)—Figure 5b; ε: 6.54 × 10^−11^ F/m (393 Hz), 3.28 × 10^−11^ F/m (21 kHz), 6.59 × 10^−11^ F/m (435 kHz), 1.70 × 10^−11^ F/m (761 kHz)—Figure 6) at different frequencies in the measured data of the sample. From Equations (2), (4), and (5), it is clear that the values of capacitance *C* and dielectric constant/permittivity *ε* have a negative correlation with the angular frequency (*ω* = 2*πf*)/frequency *f.* The slope of log *ε* at the range of frequency 1 to 1000 Hz was found −1.12 (Figure 6). The theoretical value of this slope is −1 (i.e., Equation (5)) and, thus, our measurement results of the dielectric properties of granite show the same trends as the theoretical analysis of dielectric materials. However, after 1000 Hz of frequency, the values of *ε* started to possess peak values, which indicate the dielectric relaxation in the granite, and as Maxwell’s equations on electromagnetics cannot explain the dielectric relaxation of composite materials, our simulation works would not be able to simulate some of the measured data obtained in that frequency range.

### 3.3. Electrical Properties of a Void (Pore in Mineral/Rock)

Two parallel plate electrodes of aluminum were used to measure the *I–V* properties of the air to understand the electric and dielectric behaviors of voids (pores) in granite. The diameter of the electrodes was 4 mm, and the gap was 0.2 mm for the measurement. An ultra-high resistance meter (R8340A, ADVANTEST, Chiyoda ku, Japan) was used to apply the input voltage and measure the electric current. Figure 7 shows the *I*–*V* properties of the air. The current at low-applied voltages was almost negligible, about 150 pA from 5 V to 669 V. On the other hand, at 670 V the current rapidly increased to 0.1 mA, and at 680 V the dielectric breakdown occurred. Thus, the dielectric-breakdown voltage of air was found from this measurement at 680 V/0.2 mm (3.4 × 10^6^ V/m). The dielectric-breakdown voltage of air depends on the air pressure, temperature, etc., and in general, it is known at 3.0 × 10^6^ V/m (3.0 kV for 1 cm of gap length), thus, our result of air breakdown was very similar to other reports [13,20]. From our measurement results, it is clear that the conductivity of a void (pore) is a voltage-dependent parameter.

### 3.4. Equivalent Circuit Models for Minerals

The electrical conductivity *σ* in minerals depends on the temperature. The relationship between the temperature and electrical conductivity *σ* is expressed by the following equation [11]:(6)$$\sigma =Aexp\left(\frac{-B}{k_{b}T}\right)$$
where *A* and *B* are constants, *T* is the absolute temperature, and *k_b_* is Boltzmann’s constant (8.618 × 10^−5^ eV/K). The constants *A* and *B* for each mineral in the granite, and the relative permittivity *ε*_r_ used to calculate the capacitance are shown in Table 1. In this study, we used an absolute temperature of 293.15 K, which is considered as room temperature (i.e., 20 °C). As shown in Figure 2b, two resistive elements were placed vertically in one cube, and the value of one resistive element was *R*/2. The resistance values *R* of the minerals were calculated using Equation (3). In this study, the capacitance of a mineral was calculated from Equation (2). Since the value of *C* is the value of the entire mineral in the cube of Figure 2b, the value of *C* for the mineral element was 2*C*. The values of *L* and *S* in Equations (2) and (3) were *L* = 1.0 × 10^−4^ m and *S* = 1.0 × 10^−8^ m^2^ because the minerals were arranged in 0.1 mm cubes, as shown in Figure 2b.

### 3.5. Equivalent Circuit Model of Void (Pore in Mineral/Rock)

A void is a pore in a mineral. In this study, we assumed that the void has the same electrical characteristics as the air. Normally, the air does not conduct current when an electric field is applied. However, when a high voltage is applied, a dielectric breakdown can occur, and a very large breakdown-current flows rapidly. The literature indicated that the breakdown voltage of the air is 3 kV for a 1 mm gap of electrodes and the breakdown current varies up to the scale at kA [13]. Therefore, the resistance of a void (pore) can be simulated by a voltage-dependent resistance (VDR), and the capacitor is substituted to create the equivalent circuit model for voids (pores).

In this study, the granite rock was represented by 1000 finite small cubes of 0.1 mm × 0.1 mm × 0.1 mm, and the minerals were supposed to be placed in the small cubes. From the electric and dielectric properties of the minerals (i.e., the conductivity and dielectric constant/permittivity, respectively), the circuit parameters were calculated. When a void was present in a cube with a 0.1 mm edge length within the granite rock, the breakdown voltage was 340 V/0.1 mm (=640 V/0.2 mm, Section 3.3), as we determined from our experiment on the air-discharge phenomenon. The value of the capacitor used in the equivalent circuit model for a void was calculated from the size of the cube and the dielectric constant *ε* of air (i.e., 8.85 × 10^−12^ F/m). As Equation (2) is applicable for any dielectric materials that include a void (pore), this equation was used to calculate the void capacitance that was 0.885 × 10^−15^ F and was used in our simulation work. As we described before, the conductivity of air depends on the input voltage, thus, we used a voltage-dependent resistance (VDR) system which can fit with the measurement data of the *I–V* properties of air. The equivalent circuit was simulated with the LTspice circuit simulator. The *I–V* properties of air (void) were simulated for the input voltage ranging from 1 to 350 V. The *I*–*E* characteristics of the simulation data of a void (pore/air) were prepared and plotted in a double logarithmic graph. The equivalent circuit model of the void and the *I*–*E* characteristics of the void are shown in Figure 8a,b, respectively. The electrical properties of a void (see Figure 8) indicates that before the dielectric breakdown of the air, the electric current through a void can be negligible as the value of the electric current is very low (i.e., 10^−12^ A at 1 to 350 V) when comparing to that of near the dielectric-breakdown area (i.e., 10^−3^ A at 680 V) of a void. We aimed to make an equivalent circuit model for the void, which can represent the *I–V* properties in a way that could fit the dielectric-breakdown region of the void. Figure 8b compared the *I*–*E* properties of the void obtained via measurement and simulation. From the simulation results, the electric current was 20 pA for the input voltage of 10 V (i.e., 100,000 V/m), and it remained less than 10^−9^ A even at 150 V (i.e., 1,500,000 V/m). The values of the electric current were 0.1 × 10^−9^ at 50 V (i.e., 500,000 V/m) and 0.21 × 10^−9^ A at 100 V (1,000,000 V/m)). The electric current increased gradually with the increase in input voltage, as we found it was 15.7 × 10^−9^ A at 200 V (i.e., 2,000,000 V/m) and 0.172 × 10^−6^ A at 300 V (i.e., 3,000,000 V/m). Finally, the current reached 1 mA in between the input voltage of 340 V (3,400,000 V/m) and 350 V (i.e., 35,000,000 V/m). The dielectric-breakdown voltage was found in our measurement at 680 V/0.2 mm (i.e., 340 V/0.1 mm), and, thus, we confirmed the reproduction of the *I–V* characteristics of a void (air) in our simulation work. It is to be noted that, while measuring the dielectric breakdown of the void, the system was very unstable to measure, but from the simulation works, the details of the electric current were calculated.

### 3.6. Simulation Results of Granite Sample

The patterns of minerals and void distributions can be more than thousands, but to evaluate our simulation works, we created 10 distribution patterns of minerals and voids, and then 10 equivalent circuits of granite samples. In order to evaluate the simulation works, a Ltspice circuit simulator was used to simulate the *I–V* properties of the circuits (i.e., granite) under direct-current (DC) input voltages from 100 to 1000 V. They were also simulated with AC voltage to evaluate the dielectric properties of our simulation process with a vast range of frequencies (i.e., 1 Hz to 1000 kHz).

At first, we will discuss the DC properties of our simulation results. The electric current was calculated for each input voltage for each equivalent circuit (i.e., each mineral distribution in granite). From the 10 simulation results of the circuits, the mean values and standard deviation values of the current were calculated for each input voltage. The average *J*–*E* properties were calculated and plotted in a double logarithmic graph, as seen in Figure 9. The bars indicate the standard deviation of the values (average data ± standard deviation value). We also calculated the conductivity *σ* of the granite sample for both the measurement and simulation results (Figure 10a). The average values of *σ* [pS/m] was 53.5 (measurement) and 36.2 (simulation). The standard deviation of *σ* [pS/m] was 38.5 (measurement) and 2.77 (simulation). The literature value of the resistivity (i.e., the reciprocal of conductivity) of granite was in the order of 10^12^ Ω·cm (i.e., 100 pS/m in the value of conductivity) [21] which is the same as our measured and simulation values. Thus, our simulation results are considered reliable. However, a difference in the conductivity properties of granite was found between the measurement results and simulation works. As shown in Figure 11a, it is clear that the simulation values are almost constant regardless of the input voltages, while the measured values increase with the input voltage. This difference might occur from the Schottky effect observed under a HV applied voltage on dielectric materials [13]. For an electric field *E* at temperature *T*, the emission of electrons/ions can be expressed by the following equation:(7)$$I=SDT^{2}exp\left(\frac{-eØ}{k_{b}T}\right)exp\left\{\frac{e}{k_{b}T}{\left(\frac{eE}{4\pi \varepsilon }\right)}^{\frac{1}{2}}\right\}$$
where, *D* is the Dushman constant (i.e., 1.20 × 10^6^ A/m^2^ K^2^), *T* is the absolute temperature in K, *φ* is the activation energy (i.e., similar to a work function in metal), *e* is the elementary charge quantity, and the others were described before. The details will not be discussed here, however, there is a linear relationship between the natural logarithm of electric current *I* and electric field *E*^1/2^ (Equations (8)–(10)).
(8)$$\mathrm{Ln}I=K_{0}E^{\frac{1}{2\;}}+\ln I_{0}$$
(9)$$K_{0}=\left(exp/k_{b}T\right){\left(exp/4\pi \varepsilon \right)}^{\frac{1}{2\;}}$$
(10)$$\ln I_{0}=\ln\left(SDT^{2}\right)-\left(e\phi \right)$$

These relationships can be seen in Figure 10b. For the measured data, the determination coefficient was 0.9991, while the same coefficient was 0.9583 for the simulation results. It is worth noting that the Schottky effect was not considered in this simulation work. The electrical properties of the minerals did not consider any voltage-dependent resistance (VDR), but rather, they used constant electrical and dielectric properties to create an equivalent circuit. It explains why the conductivity did not change with the voltage (Figure 10a). As we used voltage-dependent resistance (VDR) for the voids (pores) that account for only 2% in granite, the voltage change had a very limited effect on the conductivity in the simulated results. However, the mean and standard deviation values of the conductivity were very close to the measured values [21]. In fact, it can be understood from the Schottky equation that the resistivity of the rock changes greatly with the temperature, suggesting that it should be considered in future simulations. Even the recognition of voids (i.e., pores in minerals) and considering them into the equivalent circuit accurately may simulate the HV effect on the rocks more concretely. The other paper by our group discussed this relationship (the *I–V* as a function of temperature) under the HV impulse application on granite [19].

Figure 11 shows the mean values of the capacitance *C* and dielectric loss *tanσ* calculated from our simulation works. The error bars indicate the standard deviation of the values (average data ± standard deviation value). In contrast to the measured values (i.e., Figure 3a), the simulation results showed little frequency dependence. The average value of capacitance *C* was 0.0074 pF in between the frequency region of 1 kHz and 10 kHz. The mean value of the standard deviation in this frequency range was 0.000312 pF. These values are about one-tenth of the measured values (i.e., Figure 5a). The *C*–*f* characteristics of the simulation results were different from the measured values, and the values were constant even in the low-frequency range. However, the standard deviation showed a large variation up to 1 kHz, and then it became completely constant. This result suggests that the trend of the dielectric properties could be reproduced, although the average value of *C* could not be completely reproduced. In contrast to the measured values, the simulation results for the *tanδ*–*f* relationship also showed little frequency dependence. The mean value of the loss factor *tanδ* was 0.079, and the mean value of the standard deviation was 0.032. These values were also about one-tenth of the measured values (i.e., Figure 5b). Figure 5 shows that a dielectric relaxation phenomenon occurs around 1 kHz. In the measured data, the peak values for *C* and *tanδ* were found in frequencies other than 1 kHz, but in the simulation results, it was not observed at frequencies other than 1 kHz. However, we can confirm that the variation becomes larger in the low-frequency band below 100 Hz. This may correlate with the several peaks that occurred in the measured values. From this result, the dielectric phenomenon was considered to be reproduced, although the average value of *tanδ* could not be reproduced completely. It is worth noting that, unlike the measured data, this simulation work did not consider the dielectric relaxation phenomenon in the consideration, thus, the values of the simulated dielectric properties (i.e., *C*, *tanδ*) differed from the measured data. On the other hand, the general trend of the dielectric properties was similar between the simulation and measurement data.

The dielectric constant was calculated from the simulation results of each of the 10 patterns of equivalent circuit simulation, and the average value was calculated. Figure 12 shows the comparison between the calculated results and the measured data. As shown in Figure 12, the dielectric relaxation occurred in the measured dielectric constant values but was hardly observed in the simulation results. However, the measured and simulated dielectric constant values were close to each other in their magnitude. In addition, in the range of frequency between 1 Hz to 1 kHz, the measured values showed a straight line with a negative slope and a constant value after 1 kHz, while the simulation results showed a slight decrease and a constant value after 1 kHz. Thus, it is clear that, considering Maxwell’s equation for the dielectric materials (i.e., most of the rocks and granite studied in this work), the dielectric behavior for a certain range of frequency (1–100 Hz for granite in this work) can be reproduced by our simulation method to some extent.

However, it is to be noted that the equivalent circuit model method has some limitations, as recognized by our simulation results of the dielectric properties and dielectric relaxations. Capacitance values, as well as dielectric parameter/permittivity, are strongly time-dependent (i.e., frequency) parameters under a certain applied voltage. However, the capacitance value of a capacitor in the equivalent circuit model is fixed, as the parameter of capacitance cannot be replaced with a time-dependent parameter in circuit simulator software. Dielectric relaxations can be observed in composite materials with different frequencies of AC properties and are also observed in our dielectric measurements of granite. With our current model, the simulation results could not imitate the dielectric relaxations. However, as the parameter *tanδ* is determined with both dielectric and electric properties (i.e., Equations (4) and (5)), a part of the dielectric relaxation properties of *tanδ* was reproduced by our results. These basic disadvantages can be solved by considering several granite samples with a more accurate knowledge of the compositions and their dielectric properties. Further experiments will be needed to improve this method, which will be the next step in our research interests.

## 4. Conclusions

In this study, equivalent circuit models of granite were developed by considering the distributions of minerals and voids in granites. In order to confirm the validity of the equivalent circuit model, we measured the electrical and dielectric characteristics of the granite sample and compared them with the simulation results. The results presented in this article are summarized as follows:(1)The calculated electrical conductivity of the granite model and the actual granite were close to each other (measurement: 53.5 pS/m, simulation: 36.2 pS/m), and the standard deviation was very small in the simulation results (i.e., 2.77 pS/m).(2)The Schottky effect was observed in the *I–V* properties of granite, and it emphasized the necessity to consider the dielectric constant in order to simulate the *I–V* properties.(3)Comparing the simulation values of *C* and *tanδ* of the granite model and the measurement data, the dielectric relaxation phenomenon was observed in both the simulation and measurement data for the *tanδ* and frequency (*f*) relationship, and their values were close to each other. However, our simulation works did not imitate the relaxation phenomenon for the *C*–*f* relationship. For the higher frequencies (i.e., larger than 1 kHz), the simulation results showed a larger dielectric constant *ε* (i.e., 10^−9^ F/m), while the measured values were around 10^−10^ F/m.

These results suggested that the simulation model provided electrical properties similar to those of the granite sample experimentally measured. We prepared equivalent circuit models to simulate granite, but the models can be applied to any rock in order to understand the electrical and dielectric phenomena in rocks. A model considering both conductivity and the dielectric constant will be useful for understanding the electric pulse comminution. Further research works will be carried out to reproduce the dielectric properties of granite with the equivalent circuit model. We are working on a more equivalent circuit method for different ores to develop this method to be more effective in understanding the electric-pulse application behaviors of different minerals, as well as ores.

## Figures and Tables

**Figure 1 materials-15-04549-f001:**
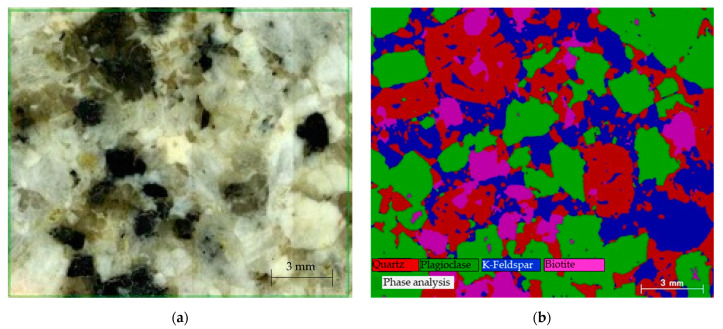
Granite sample. (**a**) Optical microscopic image. (**b**) Micro-X-ray fluorescence spectrometer image identifying quartz (red), plagioclase (green), K-feldspar (blue), and biotite (pink).

**Figure 2 materials-15-04549-f002:**
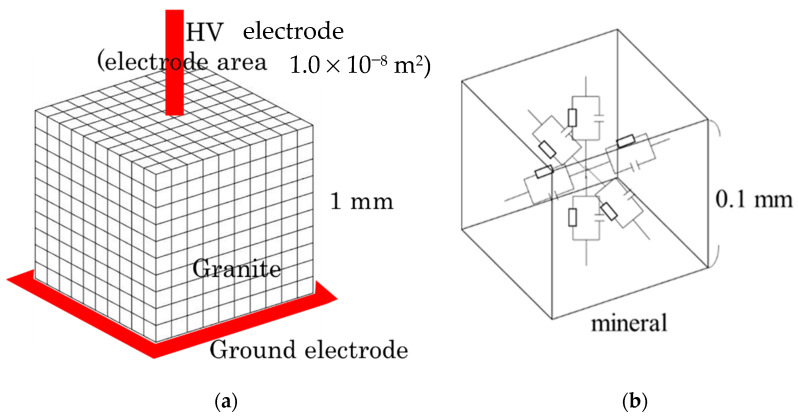
Concept of equivalent circuit model. (**a**) Granite model and (**b**) equivalent circuit model for a small, divided cube of granite.

**Figure 3 materials-15-04549-f003:**
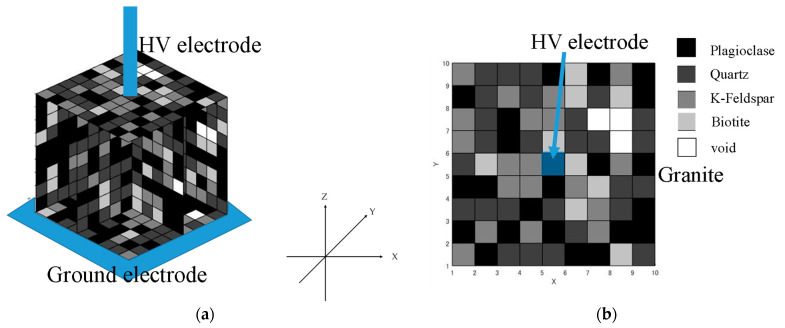
Example of mineral distribution in the simulated granite model. (**a**) Example of a granite 3D model, (**b**) XY plane of the granite model shown in (**a**).

**Figure 4 materials-15-04549-f004:**
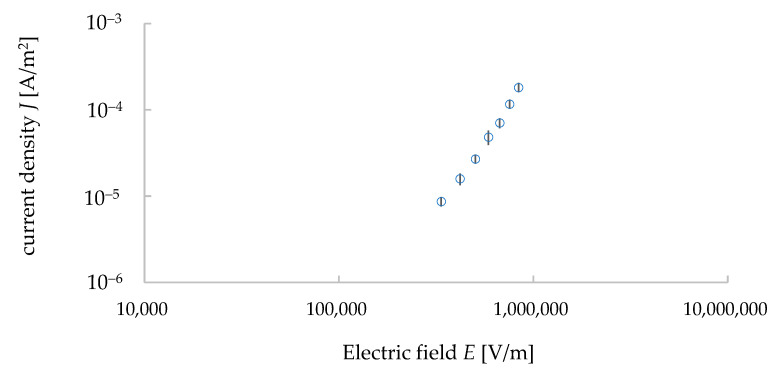
*J*–*E* relationship of granite sample (measured data) with error bars showing the standard deviation values.

**Figure 5 materials-15-04549-f005:**
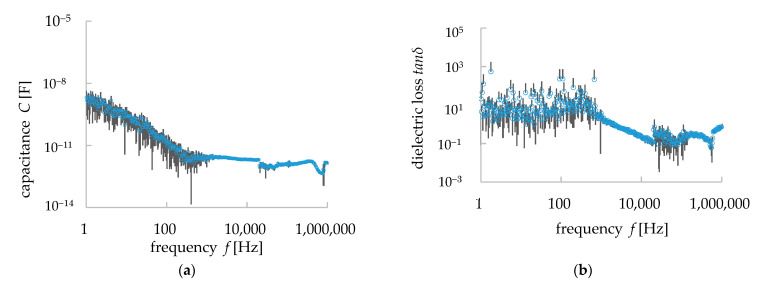
Measured dielectric properties of granite with error bars showing the standard deviation values. (**a**) *C*–*f* relationship, (**b**) *tanδ*–*f* relationship.

**Figure 6 materials-15-04549-f006:**
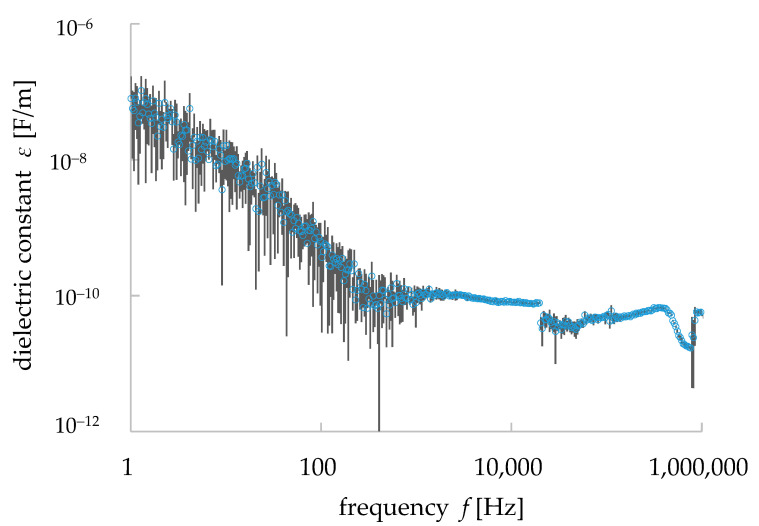
*ε*-*f* properties of granite calculated from the measured results of dielectric properties with error bars showing the standard deviation values.

**Figure 7 materials-15-04549-f007:**
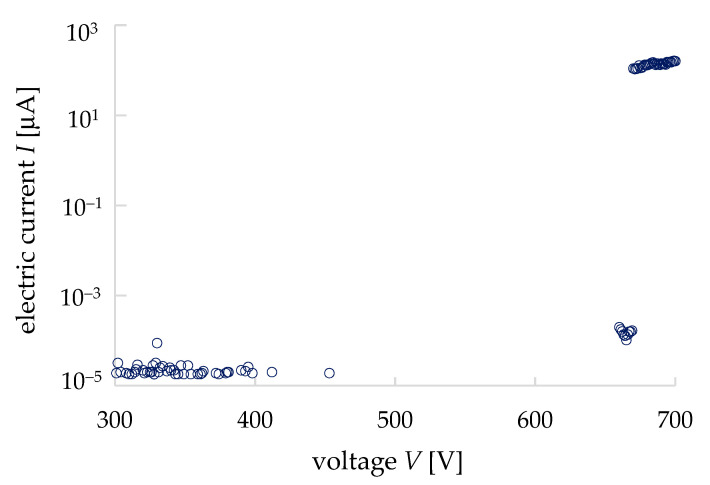
*I–V* properties of the air (i.e., similar to pores or voids in rocks).

**Figure 8 materials-15-04549-f008:**
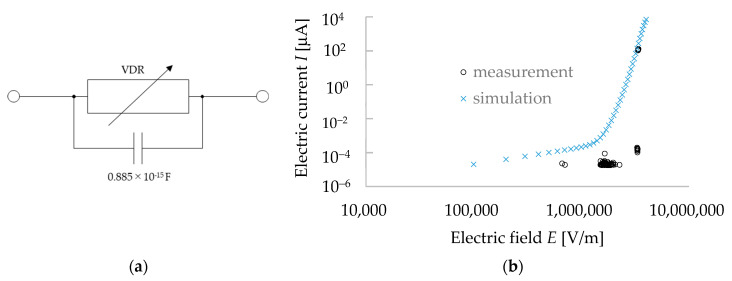
Equivalent circuit model and electrical properties of a void (pore) in minerals. (**a**) Equivalent circuit and (**b**) *I*–*E* properties of a void.

**Figure 9 materials-15-04549-f009:**
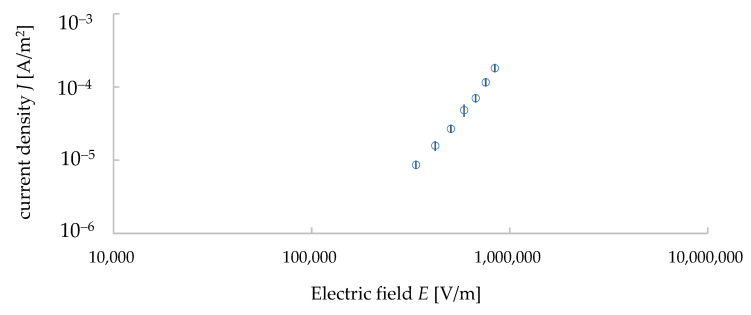
Average of simulation results of *J*–*E* for granite with error bars showing the standard deviation values.

**Figure 10 materials-15-04549-f010:**
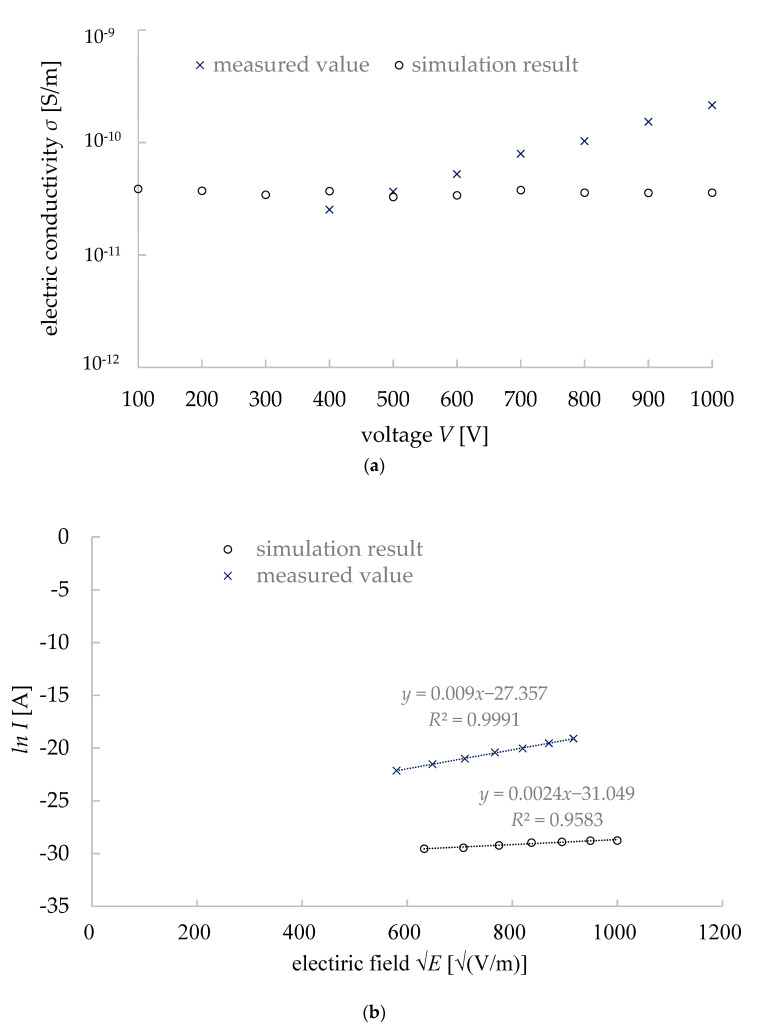
Simulation results and analysis (**a**) simulation results of conductivity of granite, (**b**) Schottky effect on granite at room temperature.

**Figure 11 materials-15-04549-f011:**
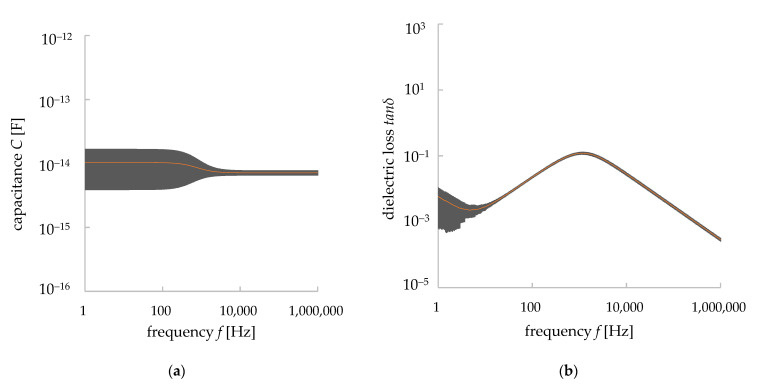
Simulation results of dielectric properties with error bars showing the standard deviation values. (**a**) *C*–*f* relationship, (**b**) *tanδ*–*f* relationship.

**Figure 12 materials-15-04549-f012:**
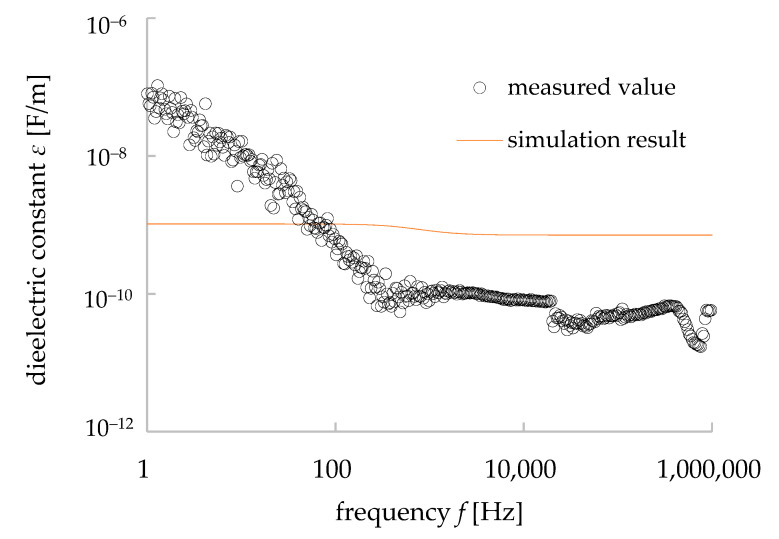
Comparison of dielectric constant with frequency in between the measurement and simulation results.

**Table 1 materials-15-04549-t001:** Parameters used in this simulation work [9,11].

Mineral	Log(*A*) [log(s/m)]	*B* [eV]	*ε* _r_
Quartz	6.3	0.82	6.53
Plagioclase	0.041	0.85	6.91
K-Feldspar	0.11	0.85	6.2
Biotite	−13.8	0	9.28

## Data Availability

Not applicable.

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
