# Peer review of "Equivalent Circuit Models: An Effective Tool to Simulate Electric/Dielectric Properties of Ores—An Example Using Granite"

_materials, 2022, doi:10.3390/ma15134549_

Round 1
Reviewer 1 Report
In this article Equivalent Circuit Models (ECM) are presented as effective tools to simulate electric and dielectric properties of ores. Granite is selected as model ore. Some simulated results are closely correlated with experimental data. The work is interesting because ECM is widely used in High Voltage (HV) engineering to simulate the behavior of HV applications for insulation/ dielectric materials. However, there are several shortcomings in the manuscript which must be addressed before its publication in Materials MDPI. My comments are as under
Abstract:
Abstract is more descriptive rather than comprehensive. It should contain some numerical values of findings of this study ( as presented in conclusions)
Introduction:
Importance of this type of study must be highlighted in the introduction. Problem statement is also missing.
Results and Discussion:
Provide references for equation 1 , 2 , 3 and 6 etc.
In section 3.2 they are talking about dielectric relaxation. Can they provide any scientific reasoning for this relaxation?
In section 3.3 they are talking about dielectric breakdown of air. Was it reversible?
Because they have assumed air to be trapped in the pores (voids) of granite. How can they correlate dielectric breakdown in air with that trapped in voids.
In section 3.4 they write "In this study, the capacitor of a mineral was calculated from equation (2)" should it not be capacitance?
The quality of some figures (like 11a) is not good. Improvement is suggested.
References:
Out of the total 29 references, 5 are authored by the corresponding author. Self-citation rule applies.
Author Response
Dear Madam/Sir,
It is our great honor for you to review our paper (materials-1696192). We would like to express our gratitude to you for your kind suggestions and comments. We have corrected our paper according to your kind suggestions. We will be highly grateful if you are kindly review it again.
Yours Sincerely,
Mahmudul Kabir
Akita University

Reviewer 2 Report
This paper presents a model based on equivalent circuits for estimating the dielectric response function of minerals based on their composition and grain structure. This model can be useful for predicting mineral properties and help extract some rarer materials using high voltage when other mechanical and chemical extraction methods may be too expensive. The paper can be published if improved. My specific comments are presented below:
The introduction sounds scattered, a compilation of earlier reports on simulating the dielectric properties of the different materials at best. The authors should present a more fundamental background and explain the hypothesis and the usefulness of this research to the broader readership in the first paragraph.
The limitation of the model should be discussed in a more focused manner. Perhaps, a paragraph at the end of the discussion should mention how the parametrization of the equivalent circuits model affects the agreement between simulation and experiment. For instance, Figure 13 shows that the simulation cannot replicate the frequency dependence of the dielectric function. Is this discrepancy inherent to the model? Or could it be improved by better assumptions like employing a more accurate chemical composition or utilizing a finer mesh than Figure 3?
The use of tildes in the title may not conform to the grammatical use case. I recommend using a colon or an m-dash
Author Response

(The authors gave the same response as above.)

Round 2
Reviewer 1 Report
The authors have complied with most of my earlier comments. The manuscript may be recommended for publication in Materials.
Reviewer 2 Report
The authors have addressed the raised concerns. The paper can be published.